# How Influenza Virus Uses Host Cell Pathways during Uncoating

**DOI:** 10.3390/cells10071722

**Published:** 2021-07-08

**Authors:** Etori Aguiar Moreira, Yohei Yamauchi, Patrick Matthias

**Affiliations:** 1Friedrich Miescher Institute for Biomedical Research, 4058 Basel, Switzerland; etori.moreira@fmi.ch; 2Faculty of Life Sciences, School of Cellular and Molecular Medicine, University of Bristol, Bristol BS8 1TD, UK; yohei.yamauchi@bristol.ac.uk; 3Faculty of Sciences, University of Basel, 4031 Basel, Switzerland

**Keywords:** influenza, capsid uncoating, HDAC6, ubiquitin, EPS8, TNPO1, pandemic, M1, virus–host interaction

## Abstract

Influenza is a zoonotic respiratory disease of major public health interest due to its pandemic potential, and a threat to animals and the human population. The influenza A virus genome consists of eight single-stranded RNA segments sequestered within a protein capsid and a lipid bilayer envelope. During host cell entry, cellular cues contribute to viral conformational changes that promote critical events such as fusion with late endosomes, capsid uncoating and viral genome release into the cytosol. In this focused review, we concisely describe the virus infection cycle and highlight the recent findings of host cell pathways and cytosolic proteins that assist influenza uncoating during host cell entry.

## 1. Introduction

Viruses are microscopic parasites that, unable to self-replicate, subvert a host cell for their replication and propagation. Despite their apparent simplicity, they can cause severe diseases and even pose pandemic threats [1,2,3]. Emerging viral infections, caused by viruses that have not been previously recorded, continue to pose a major threat to global public health [4], as it is the case for the biggest pandemic of the millennium so far, the coronavirus disease 2019 (COVID-19) caused by the severe acute respiratory syndrome coronavirus 2 (SARS-CoV-2) [5].

Entry of an enveloped virus from the extracellular environment into cells proceeds through a number of essential steps [6]. These include binding and attachment of a virus outer protein to its receptor at the cell surface, penetration of the viral particle into the cytoplasm, uncoating of the proteinaceous capsid allowing release of the viral nucleic acids into the cell cytosol, viral genetic material replication, protein synthesis, and finally new viral particle assembly and budding from the infected cell. The dissection of the molecular events and viral–host interactions that take place once the virus binds to the cell surface is essential for understanding how a particular virus infects cells. It also allows the identification of potential new targets for antivirals and therapies for blocking or controlling the infection and onset of diseases. In addition, understanding how viruses adopt or hijack cellular pathways to their advantage often leads to novel insights in the normal functioning of these pathways and is, therefore, of general interest beyond virology.

As viruses recognize target cells by first binding to cell receptors, the discovery of the virus ligands is primordial for understanding the organ tropism, the potential host diversity, and the mechanism of infection [7,8]. Enveloped animal viruses enter their host cells by membrane fusion and two pathways have been described, depending on the characteristics of the virus fusion protein. Fusion can occur at the cell plasma membrane at physiological pH [9,10,11,12,13,14,15] or within the endocytic vacuolar system where it is triggered by a low pH [16,17,18,19,20,21,22]. Virus capsid opening, the so-called uncoating, enables the virus genetic material to be released in the cytosol and get ready for replication. Our understanding of virus uncoating mechanisms has grown substantially in recent years and defines uncoating as a complex, highly orchestrated, multi-step process that relies on both viral and cellular factors.

In this review, we describe the influenza A virus (IAV) infection cycle and focus on the capsid uncoating process. We explore the latest studies that elucidate IAV capsid uncoating and the host proteins involved.

## 2. The Infection Cycle of IAV

### 2.1. Influenza Virus Structure, Proteins and Classification

Influenza viruses are orthomyxoviruses, members of the family Orthomyxoviridae, which comprises the genera Influenzavirus A, B and C, Thogotovirus, Quaranjavirus, and Isavirus [23,24]. IAV is pleiomorphic [25] meaning that viruses with varying morphologies can be produced by an infected cell. The most studied virus shape is spherical, with around 100 nm diameter; the other is filamentous, from 100 nm to 30 μm in length [26,27,28]. The filamentous morphology is typical of clinical isolates, whereas the spherical shape is common in laboratory-passaged strains [28,29]. Whilst the biological function and consequences of the viral morphology during IAV infection remain unknown, studies have shown that it has implications in transmission, host adaptation and pathogenesis [30,31,32,33,34,35,36]. IAV has eight distinct gene segments organized as a single-stranded, negative-sense RNA genome assembled into ribonucleoprotein complexes (vRNPs) that produce at least eleven proteins [37]. The segmented nature of the IAV genome has many implications, the most popular is that it provides an evolutionary benefit by enabling the virus to evolve by reassortment of gene segments between coinfecting viruses (see reference [38] for a review on this topic). Filamentous and spherical particles have their vRNPs arranged in bundle with all segments associated with the M1 from the capsid at the same end of the virus [29,39]. Figure 1 presents a scheme of the virus structure and genome.

The surface glycoproteins hemagglutinin (HA) and neuraminidase (NA) are the most abundant proteins present in the lipid bilayer envelope. Based on the antigenic properties and phylogenetic clustering of HA and NA, IAV can be classified into several subtypes. There are eighteen different HA (H1-H18) and eleven NA (N1-N11) serotypes. The relative abundance of each protein within the virus particle varies among virus subtypes and depends on the genetic background, with the HA/NA ratio being on average 4 to 1 [40]. However, for the IAV/WSN/33 (H1N1) strain it is approximately 10 to 1 [41] and for IAV/Aichi/68 (H3N2) it is 5 to 1 [39]. HA and NA play a role in the recognition and binding to the receptor in target cells and release of the virus during budding from the plasma membrane, respectively [42]. Due to the exposure at the virus surface and their biological functions, these two proteins are the major antigenic targets of neutralizing antibodies. In fact, during a natural infection the majority of antibodies will target HA, with lower amounts targeting NA or even other virus proteins [43].

IAV has two matrix proteins: M1 and M2. They are the main determinants of the spherical or filamentous virus morphology [44]. M1 is the major structural component of the virus, forming a rigid shell, the virus capsid. It acts as an adaptor between the lipid envelope and the vRNPs, besides being the driving force for virus budding [45,46,47]. A recent study solved the structure of assembled M1 within intact virus particles, gave structural insights on how M1 oligomerizes to form the capsid and how the pH change triggers the capsid disassembly [48]. Five histidine residues contributed by three sequential M1 monomers form a histidine cluster that can serve as the switch for the pH-mediated M1 disassembly [48]. M2 is an ion channel present in low amounts in the virus envelope, with approximately 20 to 60 units on each virus particle [49]. M2 forms tetrameric ion channels that open in response to the endosome low pH, allowing a proton flux into the virus. Lowering the pH of the virus interior is involved in the HA maturation by changing HA conformation from a native (nonfusogenic) structure to a fusion-active (fusogenic) [50,51,52]. The M2 protein cytoplasmic tail interacts with the M1 protein and influences virus assembly and genome packaging at the site of virus budding [53,54].

Inside the virus, each gene segment is associated with a trimeric RNA-dependent RNA polymerase complex consisting of the PB1, PB2, and PA proteins [55]. Multiple nucleoprotein (NP) molecules bind the viral RNA with high affinity and, together with the polymerase proteins, forms the vRNPs [56]. The nuclear export protein (NEP), also known as non-structural protein 2 (NS2) is found inside virus particles in low amounts where it may interact with M1 [57]. Its main function is the nuclear export of vRNPs.

The non-structural protein 1 (NS1) is abundant in IAV-infected cells but usually not detected in virus particles [58]. Nevertheless, recent studies reported that a low amount of NS1 is present in purified virus particles and suggest that NS1 can be incorporated during assembly [59,60]. Although the relevance of the presence of NS1 in the virus particles is unknown, its incorporation might have to do with its ability to associate with the IAV vRNAs and facilitate the genome packaging at the influenza budding sites [60]. NS1 is a non-essential virulence factor that has multiple functions during the viral life cycle. Its major role is to antagonize type I interferon-mediated antiviral responses [61]. NS1 also controls vRNA splicing and temporal regulation of the RNA synthesis [62,63], induces or suppresses host apoptotic responses [64,65], and has a role in strain-specific pathogenesis [66,67], among others.

### 2.2. Early Events of Influenza Virus Infection

#### 2.2.1. Receptor Binding and Envelope Fusion with Late Endosome

During the first step of IAV infection of the host cell, attachment, HA binds to the target cell via sialic acid linkages on host glycoproteins [68,69]. The sialic acid binding specificity of HA is one of the major determinants for viral tropism and host specificity; changes in key HA amino acids that control its binding specificity have been identified to contribute to the spillover of avian viruses to humans, leading to new influenza epidemics [70,71,72,73]. In general, human IAVs exhibit a strong preference for binding glycans terminating with α2,6-linked sialic acid and replicate in the respiratory tract, whereas avian IAVs have a preference for α2,3-linked sialic acid [74]. In contrast, bat IAV carries the H17 or H18 HA serotypes which cannot bind to sialic acid; rather, they require the host MHC class II proteins to infect cells [75,76]. Figure 2 illustrates the main steps of the virus life cycle.

Even though the NA is mainly recognized for its role at the virus budding stage, where it removes sialic acid bound to the newly synthesized HA and NA on nascent viruses, it has also been implicated in helping IAV to penetrate the mucus layer and get access to the receptors at the host cell membrane [77,78]. For this, NA locally cleaves sialylated O-linked glycans covering mucins and cell glycocalyx, decreasing the number of sialylated decoys and promoting the motility of IAV towards the receptors on the target cell surface [79]. The spherical IAV bound to the sialic acid-containing receptor proteins at the plasma membrane activates an internalization pathway that is by default clathrin-mediated endocytosis. In addition to this traditional route, IAV may have other entry pathways that could be dependent on the cell type. For instance, filamentous IAV enters host cells by a dynamin-independent route, using macropinocytosis as the primary entry mechanism [80]. The intact filamentous IAVs are trafficked to the acidic late-endosomal compartment within macropinosomes [81]. Similarly, spherical IAV has recently been reported to also use macropinocytosis [80]. Caveolae have already been described as an alternative route to clathrin for mediating the entry of IAV in MDCK cells [82]. By combining inhibitory methods to block both clathrin-mediated endocytosis and uptake by caveolae in HeLa cells, another study demonstrated that a non-clathrin-dependent, non-caveolae-dependent, but dynamin-dependent endocytic pathway also exists [83].

The traffic of viruses within endosomes towards the cell nucleus occurs through the cytoskeleton using actin, myosin and dynein motor protein, and microtubules (MTs) [84,85]. Polarized respiratory epithelium is the target of IAV in vivo, in which it preferentially enters the cells from the apical surface [86,87,88]. However, most molecular studies on the virus entry have been carried out using non-polarized cell lines. There are significant differences between polarized and non-polarized cells regarding receptor distribution, cytoskeletal structure and the mechanism of endocytosis [85,89]. For instance, IAV seems to depend much more on the actin dynamics in polarized than non-polarized cells [90]. Following infection, the cytoskeleton undergoes structural reorganization and the endosomes harboring viruses travel in a retrograde traffic towards the microtubule organizing center (MTOC), in close proximity to the cellular nucleus [91]. The kinetics of virus-containing endosomes vary according to the cell type and the virus subtypes. It has been reported that IAV/X31 strain can be found in early endosomes, marked by early endosomal autoantigen 1 (EEA1) [92], Rab5 [93,94], and rabenosyn-5 (Rab5 effector) [95,96], around 5 min after adsorption in dendritic cells [97]. While in Chinese hamster ovary CHO cells, the same virus already fuses its envelope with late endosomes, usually marked by lysosomal-associated membrane protein-1 (Lamp1) [98] and Rab7 [93], in the perinuclear region only 8 min after binding [84]. For IAV/WSN/33 in human lung epithelial A549 cells, co-localization of virus proteins and early endosomes peaked at 45 min whereas co-localization with late endosomes only at 120 min [99].

Upon entry through the endocytic pathway, HA only reaches a fusion-competent form when the virus has trafficked beyond early endosomes. This happens because a progressive pH drop by endosomal acidification is needed for HA conformational changes prior to fusion of the envelope and endosome membranes. The acidification of endosomes occurs during their maturation and the M2 proton channel in the virus envelope mediates the flux of protons into the IAV particle upon acid activation (pH ≈ 6) [100]. The drop of pH in endosomes has at least two main functions during early IAV stages. First, as mentioned above, the low pH in late endosomes triggers a conformational change in the HA glycoprotein that exposes a fusion peptide. Second, it strips away the M1 matrix protein from the capsid during uncoating.

#### 2.2.2. IAV Capsid Uncoating

The different steps of virus uncoating are regulated by cellular cues which come from cellular receptors, enzymes, and small chemicals including ions [101]. Here, we summarize the IAV uncoating process, and detail it further to discuss the role of host proteins in IAV uncoating (see Section 3). Uncoating refers to the series of events that alter the viral core structure leading to disassembly which is essential for the release of the viral genomic segments into the cytosol. It is a continuous and dynamic event that begins inside acidic endosomes and is completed in the cytosol by multiple host proteins that interact with viral core components. Due to the methodological limitations for identifying and visualizing all the molecules playing a role during uncoating, this process has remained relatively poorly studied. Yet, recently some important cellular proteins have been described to be involved in this process. Histone deacetylase 6 (HDAC6), epidermal growth factor receptor pathway substrate 8 (EPS8), transportin-1 (TRN-1 or TNPO1) have all been shown to be involved in the IAV core uncoating [102,103,104] and their role for other viruses begins to be examined. Catalyzing the capsid opening for a fast genome release may decrease the probability of the virus genetic material being degraded by cellular RNases as has been shown for iflaviruses, positive-strand RNA viruses from the family Iflaviridae (order Picornavirales) [105]. After release of the viral RNPs into the cytosol in the proximity of the cell nucleus, the vRNPs are imported into the nucleus through nuclear pore complexes using the nucleoprotein (NP) nuclear localization sequence (NLS) motifs and the importin α/β-dependent nuclear import pathway.

### 2.3. Nuclear Import, IAV Genome Replication, and vRNP Export

As mentioned above, the IAV genome has eight segments of single-stranded negative-sense RNA, each of them is transcribed in the nucleus of the host cell [106]. Three viral RNAs types are synthesized: viral mRNAs of positive sense (mRNA), viral genomic RNAs (vRNA) of negative sense, and complementary RNAs (cRNA) of positive sense. The vRNAs are bound by a RNA-dependent RNA polymerase, forming a viral ribonucleoprotein (vRNP) complex [107]. Although replication is a primer-independent process, during transcription of a viral mRNA the viral RNA polymerase relies on host capped RNAs as cap-donors [108]. As IAV vRNAs are of negative sense, in order for the genome to be transcribed it first must be converted into a positive sense RNA that serves as template for the production of new viral RNAs [59]. The cRNA is the replication intermediate, a full-length complement of the vRNA that works as a template for the synthesis of new copies of vRNA [107]. The formation of new vRNP complexes results from the binding of newly synthesized subunit proteins (PB1, PB2 and PA) and NP proteins to the vRNAs. For further details, the reader is referred to excellent recent reviews [109,110].

The influenza virus infection leads to a slowdown in the synthesis of cellular proteins, this phenomenon is known as cell shutoff [111,112,113,114]. The synthesis rates of vRNAs and proteins reach a maximum within the first few hours after infection before dropping [115]. One study reported the production of most viral proteins to peak in the first 8–12 h after infection [114]. The synthesis of the IAV proteins is regulated at the transcriptional level, and the synthesis rate and accumulation level of the mRNAs differ considerably among the eight RNA segments [116,117]. For instance, there is an early production of proteins such as NS1 and NP and a delayed synthesis of M1 [114,118].

Nuclear export of vRNPs is mediated by the cellular protein Crm1, or exportin 1, a member of the importin β family, and putatively by the viral protein NEP/NS2 [119,120,121,122,123]. The nuclear import relies on importin α, which acts as an adaptor between importin β and NLS-cargos [124]. M1 shuttles between the cytoplasm and the nucleus and has important functions in both compartments. In the nucleus, M1 proteins attach to vRNPs forming M1-vRNP complexes and participate in transport of vRNPs to the cytosol [125]. The NEP/NS2 protein contains a highly conserved nuclear export signal (NES) motif in its amino-terminal region and mediates the nuclear export of vRNP-M1-NS2 complexes [126]. The export of vRNP is impaired when cells are infected with a recombinant virus that cannot express NS2 or have mutations in the NS2 NES [127]. Cytoplasmic M1 proteins inhibit the nuclear import of vRNP complexes [128,129], and newly synthesized vRNPs associated with M1 protein are unable to re-enter the cell nucleus [128].

### 2.4. Late Events of IAV Infection

#### Trafficking of the vRNPs and IAV Proteins to the Cell Plasma Membrane, Genome Packaging and Virus Budding

Newly synthesized NP and the viral polymerase proteins (PB1, PB2 and PA) form a complex with vRNA, and the formed vRNP is transported to the plasma membrane or to the apical site of polarized epithelial cells for genome packaging and virus budding [130]. Similar to what happens during viral entry, the viral egress pathways depend on the cytoskeleton, transport vesicles, and motor proteins [131]. After nuclear export, vRNPs can be found colocalized with microtubules and concentrated at the MTOC [132]. The small GTPase Rab11 mediates the transport of the vRNPs across the cytoplasm to the viral budding sites at the plasma membrane [133,134,135,136,137]. It mediates the docking of a single vRNP or vRNP sub-bundles to recycling endosomes close to ER exit sites through direct or indirect interaction of its active GTP bound form with the viral polymerase complex proteins, taking the form of liquid viral inclusions [138,139,140,141,142]. Transmission electron tomography of budded IAV virions shows a distinct organization of vRNPs in which a central segment is surrounded by seven different segments of various lengths [143].

In addition to the vRNPs, IAV structral proteins (HA, NA, M1, M2) must be transported to the plasma membrane. Both HA and NA have been shown to possess apical determinants in their transmembrane domain [144,145]. HA, NA, and M2 transmembrane domains contain specific sorting signals that promote their association with sphingoglycolipid rafts at the plasma membrane [146,147]. While the transport of the IAV membrane-associated proteins has been well characterized, the mechanism by which other viral core proteins are transported to the budding sites is vague. M1 is synthesized on free cytosolic polyribosomes and may possess apical determinants or diffuse to the assembly site, or a combination of these pathways. It was shown that the M1 associates with HA and NA at the budding site and only a small fraction of the cytoplasmic M1 associates with cellular membranes in the absence of another viral protein [148]. In contrast, another study found that M1 hast the ability to associate with the membrane independent of the viral glycoproteins [45]. It is also possible that M1 may be able to associate with membranes through electrostatic interactions [149].

Virus assembly is coordinated by M1, which binds to all viral components and the plasma membrane. Interactions of M1 with other M1 proteins, vRNPs, HA and NA facilitate concentration of viral components and exclusion of host proteins from the budding site. M1 also interacts with the cytoplasmic tail and transmembrane domain of the glycoproteins HA and NA and with M2, functioning as a bridge between the viral envelope and the vRNPs [150,151]. Virus bud formation requires membrane bending at the budding site. A combination of factors including the increased concentration of viral proteins and the interaction of M1 with the viral glycoproteins, M1-M1 and M1-vRNPs play an important role for triggering virus budding [151]. Asymmetry of the lipid bilayer in lipid raft is likely to cause a curvature of the plasma membrane at the assembly site leading to bud formation. The matrix M2 transmembrane protein further facilitates virus release from the infected host cell. M2 is able to both contribute to curvature induction and also sense curvature to line up in manifolds where local membrane line tension is high [152]. During viral budding, vRNPs with their polymerase-binding ends at the budding tip are oriented in a parallel or antiparallel fashion [153]. Eventually, fusion of the opposing membranes leading to the closure of the bud will take place and newly formed viruses will be released into the extracellular environment.

## 3. In Focus: Influenza Virus Capsid Uncoating

### 3.1. Involvement of Ubiquitin Chains in Influenza Virus Uncoating

Ubiquitin (Ub) is a small 76 amino acids protein with a molecular mass of about 8.6 kDa. It participates in multiple cellular signaling pathways, that are usually involved in the regulation of protein function and homeostasis. The ubiquitin proteasome system (UPS) forms a cellular machinery for the degradation of unwanted proteins. The aggresome processing pathway (APP) is an alternative system, whereby misfolded proteins are accumulated before being degraded by autophagy. Ubiquitination and ubiquitin-like modification is usurped by many viruses to establish infection, and IAV uses ubiquitin-enhanced viral uncoating mechanisms [104,154]. How the APP facilitates efficient IAV uncoating is described below.

Protein ubiquitination is a post-translational modification in which there is an addition of a Ub molecule to one or more sites, most frequently lysine residues of a target protein. Proteins can be monoubiquitinated or poly-monoubiquitinated if they have one or more Ub molecules, respectively. In addition, Ub monomers can be connected to each another forming chains of varying lengths, linkages, and structures.

Protein ubiquitination involves a series of cellular enzymes in cascade. As depicted in Figure 3, ubiquitination starts with the Ub-activating enzyme E1, followed by the Ub-conjugating enzyme E2 and by the Ub ligase E3, which form an isopeptide bond between the carboxyl terminus of Ub and the amino group of a lysine residue on the target protein [155]. E3 ligases determine the substrate specificity of the cascade by the covalent attachment of Ub to substrate proteins, but the E2-conjugating enzyme can also play a role in the substrate selection [156]. Ub is often linked to substrates as polymeric chains that vary in both linkage and length, with important consequences for their function [157]. Ub itself contains seven lysines (K; K6, 11, 27, 29, 33, 48 and 63), all of which can be used by the Ub ligases to generate the different types of chains on the target proteins [158]. The K48-based linkages lead mainly to the proteasome-mediated degradation of the ubiquitinated protein, while K63-based Ub chains control primarily protein endocytosis, as well as trafficking and enzyme activity [159,160,161,162]. K63 is also a signal for targeting misfolded proteins to the APP [163]. Free poly-Ub chains, referred to as unanchored Ub chains, arise when deubiquitinases (DUBs) remove a chain from a protein, or they can be generated through E1/E2/E3 cycles [164]. In contrast to Ub chains bound to target proteins, unanchored Ub chains have a free C-terminus which can be bound efficiently by a conserved zinc finger domain found in the DUB isopeptidase T or in HDAC6 [165,166].

Ubiquitination has a vital role in regulating a wide variety of processes in eukaryotes through multiple mechanisms, including protein degradation, protein trafficking, gene expression, DNA repair, control of the cell cycle and signaling [167,168,169,170,171]. The versatility of the Ub system in regulating protein function and cell behavior makes it a particularly attractive target for pathogens such as viruses [172]. Ub was thought to be exclusively a cellular protein until a report described a modified form in baculovirus particles [173]. Similarly, host Ub was reported in purified vaccinia virus and herpes simplex virus particles [174]. Ub was also identified by proteomics in filovirus, such as purified Ebola and Marburg viruses [175]. More recently, liquid chromatography mass spectrometry proteomic analyses revealed a great variety of host proteins in purified extracellular viruses [176]. Among these proteins, Ub (polyubiquitin B and C) was found in Ebola Zaire, Marburg Lake Victoria, HIV-1, moloney murine leukemia virus (MLV), herpes simplex type-1 (HSV-1), vaccinia virus (VACV), human cytomegalovirus (HCMV), vesicular stomatitis virus (VSV), respiratory syncytial virus (RSV) and IAV. Ub has also been reported in HIV-1 cores [177,178]. The importance of these proteins in the different steps of the virus life cycle is unknown but Ub is long speculated to participate in virus uncoating and replication of particles [174].

Unanchored Ub chains within the IAV structural core are exposed following virus envelope and endosome fusion at late endosomes close to the nuclear periphery [104,179]. Unanchored Ub chains can also be produced by DUBs that cleave off ubiquitin chains from substrates targeted to the proteasome. One example of such a DUB is Poh1, a proteasome-associated DUB that generates K63-linked unanchored ubiquitin [180,181]. The interaction of unanchored Ub chains with HDAC6 and the interaction of HDAC6 with motor proteins in microtubules and actin filaments generates physical forces that catalyze the dissociation of the capsid M1 layer.

### 3.2. The Role of HDAC6 in Influenza Virus Uncoating

Histone deacetylases (HDACs) are enzymes that catalyze the removal of acetyl groups from modified lysine residues of histone and non-histone proteins and several classes of mammalian HDACs exist [182]. Class I HDACs are 400–500 amino acids long and include HDAC1, HDAC2, HDAC3 and HDAC8. Class II HDACs are approximately 1000 amino acids long; class IIa comprises HDAC4, HDAC5, HDAC7 and HDAC9, and class IIb comprises HDAC6 and HDAC10 [183,184,185]. The class III HDACs, also known as the sirtuins (SIRT1–7), are the silent information regulator 2 (Sir2) family of proteins and have a size ranging from 300 to 750 amino acids [186,187]. Despite the name, HDAC function is not limited to histone deacetylation and the regulation of gene transcription. HDAC6 localizes mainly in the cytosol and targets proteins through one of its deacetylase domains, CD1 or CD2, or its Ub-binding zinc finger domain, ZnF. In humans, HDAC6 contains a Ser Glu-repeat domain (SE14), which acts as a cytoplasmic retention signal and mediates its stable anchorage in the cytoplasm [188] where it deacetylases tubulin [189,190,191], heat shock protein 90 (Hsp90) [192,193,194], β-catenin [195,196], cortactin [197], MYH9, Hsc70, DNAJA1 [198] or the DEAD box RNA helicase 3, X-linked (DDX3X) [199]. HDAC6 has been associated with carcinogenesis, neurodegenerative diseases and inflammatory disorders, and has been exploited as a therapeutic target for pharmacological intervention [200,201,202,203,204,205,206,207,208].

HDAC6 has also been identified to have antiviral effects which have been linked to its enzymatic activity. It was found to inhibit IAV release by downregulating the trafficking of viral components to the plasma membrane via acetylated microtubules [209]. Overexpression of HDAC6 in cells leads to diminished viral budding due to tubulin deacetylation [210]. More recently, it was shown that HDAC6 regulates viral sensing by deacetylating retinoic acid inducible gene I, RIG-I [211], a key cytosolic sensor that detects RNA viruses through its C-terminal region and activates the production of antiviral interferons (IFNs) and pro-inflammatory cytokines. RIG-I is thought to be the most important sensor of IAV by binding to the virus genomic panhandle promoter region [212,213]. HDAC6 transiently binds to RIG-I and removes lysine 909 acetylation in the presence of viral RNAs, thus promoting RIG-I sensing of viral RNAs. Thus, HDAC6-mediated RIG-I deacetylation is critical for efficient viral RNA detection and IFN production. HDAC6 also acts as a negative regulator of IAV infection by deacetylating lysine 664 of the polymerase complex PA subunit, thereby restricting vRNA transcription and replication.

HDAC6, through its ZnF domain, associates with unanchored Ub chains [165,214,215]. As shown in Figure 3, Ub chains can be generated through E1/E2/E3 cycles and unanchored Ub chains are present in monoubiquitin, or ubiquitin derived from proteasomal degradation or the catalytic action of DUBs on pre-existing Ub chains. HDAC6 ZnF binds Ub with high affinity by recognizing its unanchored C-terminal sequence (-RLRGG-COOH) [216,217] and can recruit misfolded proteins with entangled Ub chains. In addition, HDAC6 interacts with the motor protein dynein and dynactin, the protein complex that links cargo to dynein. In this way, HDAC6 acts as a scaffold that mediates the transport of misfolded protein aggregates along microtubules and promotes formation of the aggresome compartment, which is a crucial pathway to attenuate misfolded protein-induced stress.

Inflammasome complexes are formed in response to pathogen-associated molecules and, for NLR family pyrin domain-containing protein 3 (NLRP3)- and pyrin-mediated inflammasomes, their assembly and downstream functions occur at the MTOC [218]. Similar to the formation of aggresomes, HDAC6 ZnF is required for the interaction with NLRP3 and pyrin inflammasome components and transport of these proteins using the microtubule retrograde transport by dynein for their activation in macrophages [218]. Given the importance of HDAC6 for viral uncoating, one might wonder how formation of novel IAV particles can take place in cells that contain HDAC6. The observation that in IAV-infected cells the C terminus of HDAC6 (encompassing the ZnF domain) gets cleaved off by Caspase-3 at late stages of the infection may help to solve this conundrum [219].

Ablation of class I HDACs in mice is lethal or leads to severe physiological dysfunction [220,221,222]. In contrast, mice lacking HDAC6 are viable and develop normally, despite having elevated tubulin acetylation in multiple organs [223]. The role of HDAC6 for efficient IAV uncoating was discovered by the observation that in mouse embryonic fibroblast cells lacking HDAC6, the IAV endocytic uptake, acid-induced HA maturation or fusion of virus envelope and late endosome were not affected. In contrast, virus capsid uncoating and the nuclear import of vRNPs were reduced in these cells in comparison to the wild type [104]. During infection, virus proteins and RNAs are detected by pathogen recognition receptors (PRR). This activates protein kinase R (PKR) that mediates the phosphorylation of eukaryotic initiation factor-α (eIF2α) on serine 51 to initiate the assembly of virus-induced stress granules concomitant with repression of cellular proteins translation [224]. By mimicking a misfolded protein aggregate, IAV hijacks the APP to its benefit [104]. The role of HDAC6 and Ub for IAV capsid uncoating can be visualized in Figure 4.

Moreover, HDAC6 knockout mice intratracheally infected with IAV showed reduced lung viral titers compared to wild type mice, whereas the antiviral immune responses were comparable ([104] and our unpublished results). This showed that the pro-viral role of HDAC6 ZnF domain during IAV uncoating influenced the infection outcome. In contrast, another recent study in which another strain of HDAC6 knockout mice was infected with IAV showed them to be more susceptible to PR8 H1N1 infection than their wild type counterpart [225]. In this work, it was argued that the absence of HDAC6 leads to a blunted innate response and concomitantly increased susceptibility of mice to IAV infection. The reason for these differences is not known but might possibly reflect the presence of different microbiomes in the two strains. In the future, targeted mutation of the HDAC6 ZnF or CD in mice is desired to fully understand the in vivo pro-viral and antiviral effects of HDAC6, respectively.

### 3.3. Epidermal Growth Factor Receptor Pathway Substrate 8

The epidermal growth factor receptor pathway substrate 8 (EPS8) is an adaptor protein involved in signaling via the epidermal growth factor receptor (EGFR) [226,227]. EPS8 also directly binds to actin filaments controlling the rate of polymerization and depolymerization by capping the fast-growing ends of actin filaments [228,229,230,231]. EPS8 regulates intracellular trafficking of membrane receptors through its direct interaction with the GTPase-activating protein RN-tre, which controls the activity of Rab5, or by interacting with the clathrin-mediated endocytosis machinery.

A screen for host factors involved in IAV infection by correlating WSN H1N1 infectivity with gene expression profiles of 59 distinct cell lines identified EPS8 as the highest confidence pro-viral host gene [102]. Knocking out EPS8 in human A549 lung cells decreased viral titers in the infected-cell supernatant by 10-fold in multicycle replication assays. The loss of EPS8 did not affect virus attachment, uptake or fusion. EPS8 physically associates with incoming virus components possibly through interactions with NP, the viral polymerase, M1, or bridged by other cellular uncoating factors (Figure 4). EPS8 might interact with vRNPs by binding to NP as the viral nucleoprotein specifically co-precipitated with EPS8. Additionally, the import of vRNPs was significantly delayed in cells lacking EPS8 in comparison to WT cells, leading to a reduction in viral gene expression [102]. EGFR signaling, which promotes IAV entry [232], was unaffected by EPS8 depletion [102]. Although mechanistic details are missing, one can speculate that EPS8 regulates actin filaments to enhance IAV uncoating [102].

### 3.4. SPOPL/Cullin 3 Ubiquitin Ligase Complex and EPS15

The maturation of late endosomes/multivesicular bodies entails the spatial and functional separation of the organelles from early endosomes, preparing them as a feeder pathway to lysosomes [233]. Cellular processes that promote endosome maturation play a critical role in influenza uncoating. Cullin3 (CUL3)-based E3 ubiquitin ligases regulate endocytic trafficking of cargo to lysosomes and endosome maturation. Transfer of cargo from early endosomes to lysosomes depends on an endosomal maturation process regulated by a variety of protein- and lipid-based events. They include a small GTPase Rab5-to-Rab7 switch, a PtdIns(3)P to PtdIns(3,5)P2 conversion, and changes in the luminal ion concentrations, such as decrease in pH and increase in K^+^ concentration [233,234]. Using a siRNA screen against 130 human Bric-a-Brac/Tramtrack/Broad (BTB) domain proteins in A549 cells, it was found that the Speckle-type POZ protein-like (SPOPL) was crucial for EPS15 ubiquitination by the Cullin RING E3 ubiquitin ligase 3 (CRL3)^SPOPL^ complex. EPS15, an endocytic adaptor that associates with ESCRT0 proteins HRS and STAM, was necessary for endosome maturation and IAV capsid disassembly [235]. The depletion of SPOP and SPOPL gave a similar phenotype to Cul3 depletion [236], showing retention of viral components in the endocytic system and inhibition of infection. Ubiquitin-modifying enzymes that regulate endosome maturation play a yet incompletely understood but important facilitator role in the successful uncoating of IAVs.

### 3.5. Transportin 1

In eukaryotic cells, transcription and translation are physically separated by the nuclear membrane; transcription occurs only within the nucleus, and translation occurs only outside the nucleus in the cytoplasm. The nuclear membrane, also known as nuclear envelope, is a phospholipid bilayer that encloses the cell nucleus and is penetrated by nuclear pore complexes. Small molecules (usually less than 60 kDa) diffuse freely through the nuclear pores [237,238]. Alternatively, proteins may shuttle between the cytoplasm and the nucleus in an active way that is mediated by nuclear localization signals (NLSs) or nuclear export signals (NESs). In this way, larger molecules are selected by nuclear transport receptors (also called karyopherins) that carry their cargoes from one compartment to the other by crossing the nuclear envelope at the level of the nuclear pore complexes [239,240]. Importins mediate the nuclear import of cargos and transportin 1 (TNPO1, also known as importin-β2, KPNB2) is one of the best-characterized nuclear import receptors [241].

Many viruses depend on nuclear proteins for replication and their viral genome must enter the nucleus of the host cell. This is the case of most DNA viruses and some RNA viruses, including orthomyxoviruses and retroviruses. Therefore, it is expected that the life cycle of these viruses is dependent on transporters (e.g., importins, exportins, transportins) and regulators (e.g., Ran GTPase). Great effort has been put on deciphering viral nuclear transport mechanisms [242,243,244,245]. In the context of IAV infection, a study using RNAi for targeting, among other proteins, nuclear pore proteins identified TNPO1 as an important host factor involved in uncoating. Depletion of TNPO1 in different cells reduced the number of infected cells and the production of new viruses [103]. It was shown that TNPO1 was important not only for the vRNPs nuclear import, but also for the M1 uncoating and vRNP debundling in the cytosol [103]. Moreover, the role of TNPO1 in uncoating was associated with its recognition of a nuclear localization signal as it binds to the exposed M1 N-terminal PY-NLS motif only after capsid acidification. As shown in Figure 4, by recognizing and binding the M1 NLS, TNPO1 promotes the removal of vRNP-associated M1, which leads to dissociation of vRNPs from each other and facilitates further nuclear import by importin α and β via the classical NLS-mediated import pathway. It is noteworthy that as endosomes mature, both decrease in pH and increase in K^+^ concentration in the lumen of late endosomes take place, which is important for sufficient priming of the viral core for uncoating [234]. A high K^+^ concentration, in particular, promotes dissociation of bundled vRNPs from each other in an in vitro uncoating assay [234]. The segmented nature of IAV vRNPs not only promotes reassortment during co-infection [246] but may also allow the segments to be transported in and out of the nuclear pore individually.

## 4. Perspectives in the Field of Virus Host Interaction and Capsid Uncoating

Due to their nature, viruses need to constantly interact with their host cells. They are always trying to either counteract or exploit different cellular mechanisms and pathways to their advantage. Better understanding the molecular requirements viruses have on host cells or the immune mechanisms used by the cells to escape infection is important for the development of novel approaches to fight viral infections.

Despite the availability of licensed vaccines, IAV is estimated to be responsible for 290,000 to 650,000 worldwide flu-associated deaths annually [247] and is of major public health interest due to its pandemic potential and constant threat to animals and humans. This review focused on the IAV life cycle highlighting the interactions with the cell host proteins. Capsid uncoating is a dynamic process that has remained relatively poorly studied. However, in recent years, progress in this field has been made with the identification of cellular proteins and pathway involved in IAV uncoating.

Enveloped viruses carry several host proteins in their structural core after budding has taken place. One of these proteins is Ub, which in the form of unanchored Ub chains, can recruit cellular proteins in the infected cells, such as HDAC6. Similarly to what happens to the aggresome and inflammasome pathways, Ub chains recruit HDAC6 that acts as a scaffold protein, interacting with virus proteins from the capsid and virus genome as well as with cytoskeletal motor proteins. These interactions generate physical forces that catalyze the dissociation of the capsid M1 layer underneath the viral envelope. In parallel, EPS8, TNPO1 and possibly other cellular proteins and kinases such as G protein-coupled receptor kinase (GRK2) [248], as well as endosome maturation, together contribute to generate the cellular environment that ultimately leads to uncoating and release of individual vRNPs at the perinuclear area.

Considering that other viruses could use a similar mechanism during their virus cycle, it is important to investigate if the HDAC6-mediated APP is involved in uncoating of other enveloped viruses that also have Ub in their viral mature particles. Similar to IAV, other important viruses including HIV-1, Ebola, rabies, HSV-1, VACV, HCMV, VSV, RSV, also carry Ub in their particles [175,176,177,178]. The identification of additional host proteins in viral particles could give hints to the possible pathways used by viruses during their life cycle. Considering that host proteins incorporated by viral particles might play crucial roles, as Ub does, it is important to realize that different host cells may influence the composition of the host protein profile inside virus particles. This was reported for HIV, in which the host protein profile in mature particles was found to be different depending on the cell host from which it originated [177]. This might be even more important for viruses that transition from one host species to another during their life cycle. Arboviruses (Zika, dengue, yellow fever, chikungunya, tick borne encephalitis etc.) are examples of viruses that infect mammals and arthropods during their life cycle. IAV, Ebola and SARS-CoV are other zoonotic viruses that jump from animals to humans and can be at the origin of pandemics.

TNPO1 also plays a role in the uncoating of HIV. Similar to the mechanism of IAV uncoating, TNPO1 binds to HIV capsids, triggers their uncoating and promotes viral nuclear import [249]. Given that some of the host proteins that play a fundamental role in IAV uncoating have been since then shown to participate in the uncoating of other viruses, it is interesting to think about the potential of interfering with this step of the virus life cycle by targeting one or more of these host proteins. Indeed, targeting host processes has a potential advantage of being less likely to give rise to viral resistant variants and to be of broad use for different viruses. Studies in this direction led to the development of the only host-targeting antiviral agent among the 20 approved antiretroviral used to treat HIV patients: maraviroc, a virus entry inhibitor that targets the chemokine receptor CCR5 expressed on the surface of white blood cells [250,251]. However, apart from immunomodulators, almost all antiviral drugs currently approved or under development target viral proteins. For IAV, NA inhibitors, M2 channel blockers, and PA endonuclease inhibitors are the three classes of inhibitors approved for treatment [252,253]. The administration of inhibitors of the M2 ion channel has been discouraged by the CDC due to widespread pre-existing viral resistance among H3N2 and H1N1 strains [253]. This highlights the need for new antiviral strategies with novel mechanisms of action and reduced drug resistance potential. Thus, if one identifies host proteins that are fundamental for the uncoating or replication of multiple viruses, these would have the potential to become novel targets for a broad-spectrum antiviral drug. While the main disadvantage of host-targeted antivirals is the higher risk for host toxicity, an advantage is that the host-targets/proteins can be studied before a new virus emerges. In addition, a host-targeted approach often offers a higher barrier to the appearance of viral drug resistance [254]. The development and approval of such new antivirals could be of great use for viral pandemic preparedness and complement vaccines.

## Figures and Tables

**Figure 1 cells-10-01722-f001:**
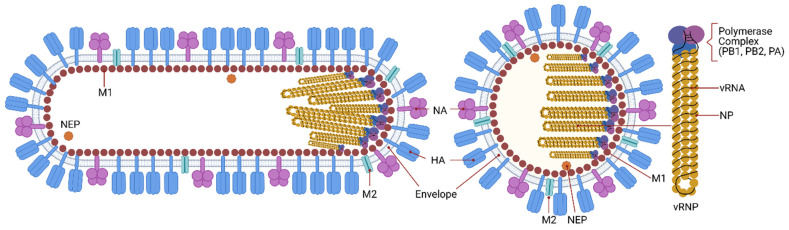
IAV structure and genome. Influenza is an enveloped virus in which structural proteins can be found associated with the virus envelope, a lipid bilayer derived from the plasma membrane of the host cell. The viral envelope contains three of the viral transmembrane proteins: hemagglutinin (HA), neuraminidase (NA), and the matrix ion channel M2. HA and NA proteins are the main proteins at the virus surface and HA is four times more abundant than NA. M2 also penetrates the envelope but represents a minor component of the envelope, with about 20 molecules per virus particle. The matrix protein M1 is found beneath the lipid membrane, and forms a rigid single-helical layer shell, the virus capsid. The nuclear export protein (NEP) is found in the interior of the virus. The IAV genome consists of eight negative-sense RNA segments that form distinct viral ribonucleoproteins (vRNPs). vRNPs are assembled as virus RNA segments where the termini of viral RNAs associate with the viral RNA-dependent RNA polymerase complex, PB1, PB2 and PA, while the rest of the viral RNAs are bound by oligomers of the nucleoprotein, NP. The virus has an asymmetric internal structure, maintained by vRNPs-vRNPs and M1-vRNPs interactions. Not shown in the figure, the interior of IAV bears a substantial number of host proteins (ubiquitin, tubulin, actin, annexin, among others). IAV is known to display a number of shapes. The spherical form of IAVs is typically about 100 nm in diameter. Filamentous forms of IAVs can be over a few μm in length.

**Figure 2 cells-10-01722-f002:**
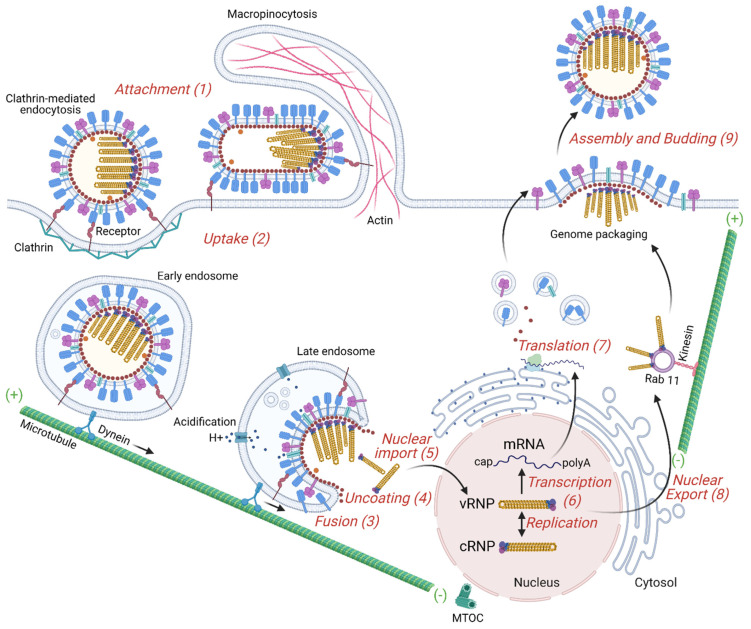
Overview of the IAV replication cycle. The influenza virus life cycle can be divided into several stages: (**1**) Virus binding to the target cell. HA binds to sialic acid found on the surface of the host cell’s membrane. (**2**) Entry into the host cell: a clathrin-mediated endocytosis or macropinocytosis takes place. Early endosome containing viruses is transported by dynein along microtubules to the perinuclear region close to the microtubule-organizing center (MTOC). (**3**) Fusion of the virus envelope with the endosomal membrane. Acidification increases progressively from endocytic vesicles to late endosomes and induces a HA conformational change to a fusion-competent state. M2, an acid-activated viral ion channel, is required for efficient viral envelope fusion with the endosomal membrane and nucleocapsid release. (**4**) Uncoating of the virus capsid by disassembly of the M1 proteins and release of the viral ribonucleoproteins (vRNPs) to the cytosol. (**5**) Entry of vRNPs into the nucleus by an active nuclear import pathway. (**6**) Transcription and replication of the viral genome. The IAV genome is composed of negative-sense strand RNAs. The genome is first converted into positive-sense RNAs, forming complementary ribonucleoprotein (cRNP) complexes, that serve as templates to produce viral RNAs. The transcription of the vRNA generates mature viral messenger RNAs (mRNAs) that have a 5′ methylated cap and a poly(A) tail. (**7**) Viral protein translation occurs by free ribosomes or ribosomes on the rough endoplasmic reticulum. Some of these proteins enter the nucleus where they assemble with viral RNAs. (**8**) Export of the vRNPs from the nucleus. vRNPs are exported out of the nucleus via the CRM1 dependent pathway through the nuclear pores. (**9**) Transport of viral components, assembly and budding at the host cell plasma membrane. Viral glycoproteins, HA and NA, associate with lipid rafts, membrane microdomains comprised of densely packed cholesterol and sphingolipids. vRNP complexes are transported as sub-bundles on Rab11 to recycling endosomes close to ER exit sites toward the plasma membrane and are incorporated as a complex of eight different vRNPs into budding viruses. Finally, the plasma membrane containing the viral structural proteins at the assembly site bends releasing infectious virus into the extracellular environment.

**Figure 3 cells-10-01722-f003:**
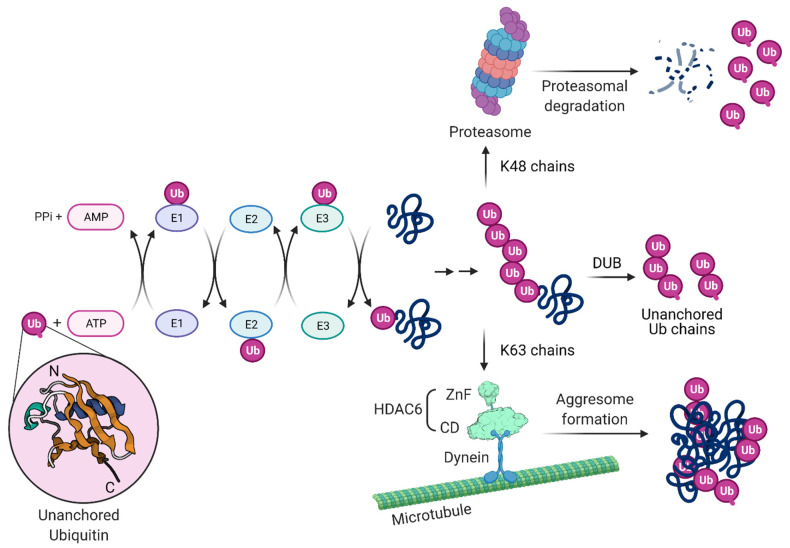
Protein ubiquitination, ubiquitin chains, proteasomal and aggresome-autophagy degradation. Ubiquitin (Ub) is a small, 76 amino acid protein present in all eukaryotic cells that plays a key role in the cellular defense mechanism by functioning as a proteolytic signal for the proteasome. The process of covalent Ub attachment to target proteins is called ubiquitination (also known as ubiquitylation). This post-translational modification forms by an isopeptide bond between the carboxyl terminus of Ub and a lysine residue on the target protein. First, Ub is covalently conjugated to the E1 (Ub-activating enzyme) in an active ATP-dependent reaction and transferred to the E2 (Ub-conjugating enzyme). The E3 (Ub–protein ligase) transfers the Ub from E2 to the target protein and determines the specificity. A monoubiquitinated protein can have the Ub chain elongated by E3 that creates Ub–Ub isopeptide bonds. Chain extension can happen through seven lysine (K) residues on Ub: K6, K11, K27, K29, K33, K48 and K63. Proteasomes recognize K48 chains leading to target protein degradation (upper part). Other lysine chains are involved in different biological functions. K63 chains do not specify degradation but usually facilitate the recruitment of other proteins in the formation of functional complexes involved in cellular signaling such as aggresome formation (lower part). HDAC6 can bind misfolded proteins entangled with Ub K63 chains and bridges to dynein motors, mediating transport to and formation of the aggresome compartment. Free poly-Ub chains, referred to as unanchored Ub chains, have been found in virus particles. Unanchored poly-Ub chains arise when a deubiquitinase (DUB) removes an intact chain from a protein, or they can be generated through E1/E2/E3 cycles. They can be recognized by HDAC6 and activate the aggresome pathway as well IAV capsid uncoating. Ub Protein Data Bank (PDB): 1UBI; NH2 and COOH termini are labeled N and C, respectively. HDAC6 zinc finger (ZnF) and catalytic domain (CD) PDB: 3C5K and 5G0I, respectively.

**Figure 4 cells-10-01722-f004:**
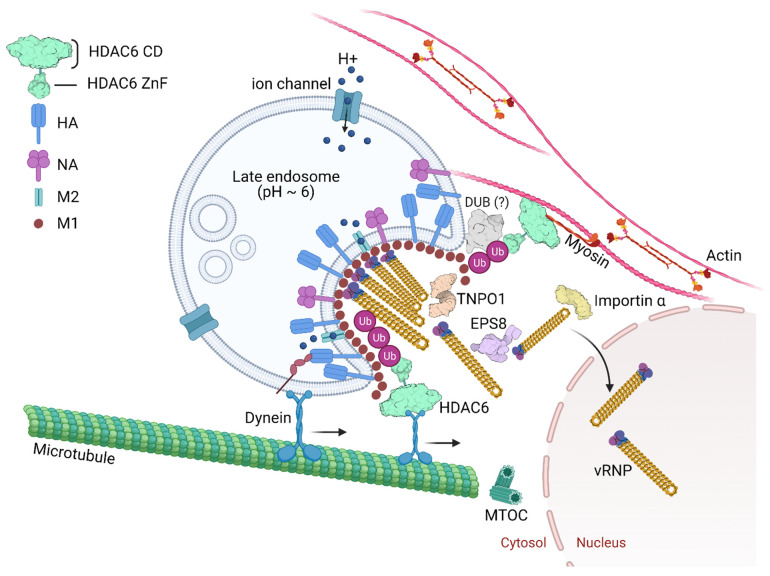
IAV capsid uncoating, genome release and nuclear import. Endosome acidification occurs progressively from the cell periphery toward the microtubule-organizing center (MTOC). Late endosomal acidification (pH~6) triggers change of the homotrimeric glycoprotein HA mediating fusion between the viral envelope and the endosome membrane. Influx of protons and efflux of potassium from the virus core happen through the acid-activated viral ion channel M2. The pH drop triggers the activation of a histidine cluster in the virus capsid, contributed by three sequential M1 monomers, and promotes the capsid disassembly. Further the vRNPs dissociate from the M1 proteins. Free ubiquitin (Ub) chains derived from virus particles activate the aggresome processing pathway (APP) and recruit HDAC6 through its Ub-binding zinc finger domain (HDAC6 ZnF). Deubiquitinases (DUBs) could be involved in unanchored Ub formation. HDAC6 binds to M1 and to NP from vRNPs. HDAC6 by a region between its catalytic domains also binds motor proteins in microtubules and myosin II in actin microfilaments generating physical forces that help dissociate the M1 proteins, disassembling the virus capsid. The epidermal growth factor receptor pathway substrate 8 (EPS8) and transportin-1 (TNPO1) interact with M1 from the capsid and vRNPs, contributing to the disaggregation of the vRNP-associated M1 and vRNP debundling in the cytosol. In this way, vRNPs are transported by importin α/β to the nucleus as individual rod-shaped structures. PDB: TNPO1 (2Z5J), EPS8 (2E8M), importin α (4B18), DUB (6K9P).

## Data Availability

Not applicable.

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
