# Peer review of "How Influenza Virus Uses Host Cell Pathways during Uncoating"

_cells, 2021, doi:10.3390/cells10071722_

Round 1

Reviewer 1 Report

Comments to Etori Aguiar Moreira et al. Host Cell Pathways Used by Influenza Virus during Uncoating 2. Here are some high level comments, as I am not particularly familiar with the low level details. Recommendation: major revision absoluetly required for ch 1, 2, 4. Revision of ch 3 is also needed. Salient points are mentioned below.

  • Title is confusing. It suggests that there is many ways to uncoat an influenza virus. Is this really true? If so, what are these mechanisms? I am not familiar with this. Plus, last sentence of abst contradicts this.
  • Abst 1st sentence: smth is wrong
  • 2nd sentence: smth is wrong. Viruses also interact with host components outside of cells.
  • 3rd sentence: what does this have to do with the above or below?
  • General impression on chapters 1, 2: poorly written.
  • Intro: first sentence is wrong.
  • Line 24: confusing
  • Line 25: not correct, acc to my little knowledge not all viruses necessarily reprogram cells. There are beautiful examples of passenger viruses out there.
  • Line 27: what is a novel pathogen? Poor language, confusing alarmistic sentence. What does COVID have to do with IAV? Needs to be spelled out or omitted.
  • Intro in general is not very useful as the reader does not find the approprite references next to specific statements.
  • Line 63: what is the point in studying a virus shape? This sounds exotic.
  • Line 102: how is M2 involved in pH regulation? Unclear to me.
  • Ref 41: what is the evidence for this, and why would this be important? Not intuitive and convincing to a naïve reader.
  • Line 125: This is striking. I looked this paper. Did they really show binding to MHCII?
  • Fig 2: Buckled microtubles is striking. What is the evidence for this? Are they involved in IAV uncoating?
  • Line 167: what is an intact filament? Microtubules?
  • Line 170: very interesting statement. What is the evidence?
  • Line 174: Is this generally true also for polarized epithelial cells?
  • Line 180: What is rabenosyn-5? Synonymous to Rab5?
  • Line 192: is M2 a transporter or a channel?
  • Line 195: I am confused. Low pH strips M1 from the capsid. What is capsid?
  • Line 211: another striking statement. I looked up the reference. But the reference refers to a picornavirus not a flavivirus. I am confused.
  • It is not clear to me what chapter 2.4. has to do with IAV uncoating, as in title of review.
  • Chapter 3: the focus on IAV is lacking, and text frequently deviates to other topics that seem not to be related to IAV uncoating.
  • Line 376: very interesting. What is the evidence that DUB is involved?
  • Line 404: is RIG-I involved in IAV, and if so how?
  • Line 460: ref 190 and ref 74 report completely opposite results with regards to the susceptibility of HDAC6 KO mice to IAV H1N1. The discrepancy between the two studies should be discussed.
  • Fig 4: where is the capsid?
  • Not clear how events described in chapters 3.3. 3.4. are connected to IAV uncoating.
  • Chapter 4 is jumpy and does not give an easy read.
  • Line 584: Is there any evidence for this?
  • Line 587: what is an emergency drug?

Reviewer 2 Report

The manuscript by Etori Aguiar Moreira et. al. untitled “Host Cell Pathways Used by Influenza Virus during Uncoating” is an interesting and detailed review. Authors presented current understanding of the influenza virus and host interactions, focusing on the capsid uncoating. The reviews cited the recent literature and is a comprehensive knowledge on the title subject. I recommended to accept manuscript after minor changes. The detailed remarks:

1/ I do not fully agree with a sentence: “The genome of IAV is encoded in eight distinct viral ribonucleoproteins (vRNPs) complexes that have negative RNA sense”. It should be rather: The IAV genome consists of eight negative-sense RNA segments that form distinct viral ribonucleoproteins (vRNPs). IAV genome (by definition) is RNA only.

2/ The role of RNA genome and consequences of its splitting to segments should be more emphasized in manuscript.

Round 2

Reviewer 1 Report

none